# Malaria prevention knowledge, attitudes, and practices (KAP) among adolescents living in an area of persistent transmission in Senegal: Results from a cross-sectional study

**Fassiatou Tairou**[1]*, **Saira Nawaz**[2], **Marc Christian Tahita**[3], **Samantha Herrera**[4], **Babacar Faye**[1], **Roger C. K. Tine**[1]

1 Department of Medical Parasitology, University Cheikh Anta Diop of Dakar, Dakar, Senegal, 2 Primary Health Care, PATH, Seattle, Washington, United States of America, 3 Clinical Research Unit of Nanoro, Direction Régionale du Centre-Ouest de l'Institut de Recherche en Sciences de la Santé, Bobo-Dioulasso, Burkina Faso, 4 Malaria & Neglected Tropical Diseases Division, PATH, Washington, District of Columbia, United States of America

* fassiatht@yahoo.fr

**Data Availability Statement:** All relevant data are within the paper and its Supporting information files.

## Abstract

### Introduction

While malaria morbidity has sharply declined in several areas in Senegal, it remains an important problem in the southern part of the country, particularly among adolescents. Understanding adolescents' knowledge, attitudes, prevention and care-seeking practices is important to inform more targeted interventions aimed at optimizing adolescents' uptake of malaria prevention and control measures. This study assessed malaria-related knowledge, attitudes, and practices (KAP) among adolescents living in a highly persistent transmission area in Senegal.

### Methods

A community-based cross-sectional survey was conducted among 391 adolescents living in the Saraya health district. A multistage random sampling technique was used to select households. An electronic questionnaire developed on Open Data Kit (ODK), was used to collect data on socio-demographic characteristics, household assets, adolescents' knowledge of malaria, as well as their attitudes with regards to malaria prevention, and care-seeking behaviors. Bivariate and multivariate analyses were performed to assess factors associated with adolescents' KAP towards malaria.

### Results

Nearly, one-third of the participants had good knowledge of malaria (34.4%) and good practice in regards to malaria preventive measures (32.8%) while 59.0% had a positive attitude and 73.8% had good care-seeking behavior regarding malaria. Multivariate analysis revealed that a primary (aOR = 5.43, p = 0.002) or secondary level of education (aOR = 10.41, p = 0.000) was associated with good knowledge of malaria transmission, signs, and

**Funding:** The study was supported by the Senegalese National Malaria Control Program, through an agreement with the Department of Parasitology and Mycology of University Cheikh Anta Diop of Dakar (# funding number: NFM 2-SEN-M-PNLP 2018 - 2019 -2020). The funder had no role in the study design, data collection, and analysis, decision to publish, or preparation of the manuscript.

**Competing interests:** The authors have declared that no competing interests exist.

prevention measures. Male individuals had lower knowledge compared to female ones (aOR = 0.40, p = 0.001). Individuals belonging to households from the highest wealth quintile were more likely to have a positive attitude towards malaria compared to those from households in the lowest wealth quintile (aOR = 3.49, p = 0.004). The odds of positive attitude towards malaria decreased among participants with koranic and primary education level, respectively (aOR = 0.14, p = 0.005) and (aOR = 0.24, p = 0.019). A positive attitude was 1.89 more likely to be (aOR = 1.89, p = 0.026) associated with good practice of prevention measures compared to adolescents who demonstrated negative attitudes. Individuals from households in the fourth (aOR = 0.42, p = 0.024), middle (aOR = 0.34, P = 0.005), and second (aOR = 0.42, p = 0.027) wealth quintiles were less likely to use malaria prevention measures compared to those from households in the highest wealth quintile.

## Conclusion

The study revealed that adolescents, generally have poor levels of malaria knowledge and low uptake of malaria prevention and control interventions. Targeted interventions for high-risk adolescents are needed, that focus on improving their knowledge of the disease and effective preventive measures, and on increasing their access to health care services and LLINs.

## Introduction

Malaria remains a severe global public health problem, despite important efforts to control the disease. Malaria cases declined globally from 238 million in 2000 to 229 million in 2019 [1], while the incidence rate decreased from 80 per 1000 population at risk in 2000 to 57 in 2019 [1]. However, the Coronavirus (COVID-19) pandemic has disrupted malaria control activities, threatening progress in the fight against the disease [2, 3], leading in 2020 to an increase in malaria morbidity and mortality worldwide, with an estimated 241 million cases and 627 000 deaths [4]. The incidence rate (per 1000 population at risk) increased from 57 in 2019 to 59 in 2020 [1]. Most of these cases occurred in Sub-Saharan Africa (SSA), with an estimated 228 million cases and 602 000 deaths in 2020 [4]. In recent years, multiple studies reported increased risk or higher prevalence of malaria among old children and adolescents across several SSA countries, including Kenya [5], Malawi [6, 7], Tanzania [8], Democratic Republic of Congo [9] and Senegal [10], suggesting a shift of the disease burden from children under five years old to older children and adolescents.

In Senegal, malaria is endemic in most of the country, with an upsurge during the rainy season [11]. Malaria control relies on prompt and accurate diagnosis, treatment using artemisinin-based combination therapy combined with preventive measures such as use of long-lasting insecticide-treated nets (LLINs), indoor residual spraying (IRS), intermittent preventive treatment in pregnant women and children. The scaling-up of these interventions has led to a substantial reduction in the malaria burden in the country. Proportional morbidity has decreased from 4.9% in 2015 to 3.3% in 2019, and proportional mortality from 3.5% in 2015 to 1.7% in 2019 [12]. However, an increase in malaria burden was observed in 2020, the proportional morbidity and mortality increased to a level close to the observed trends in 2015; respectively 3.8% and 2.1% [13, 14]; and the majority of these malaria cases occurred in the three southern regions of Kolda, Tambacounda, and Kedougou [13, 14]. In addition, the Senegalese

National Malaria Control program (NMCP) through longitudinal routine data has shown an important increase in malaria cases among older children (15 to 19 years) compared to children under-five years old. The observed upward trend in malaria risk among older children and adolescents could be due to suboptimal use of preventive measures [15, 16] and their behavioral patterns, such as staying outdoors in the evening hours when mosquitos are most active [16, 17].

Given this shift in the malaria burden, it has become increasingly important to develop strategies to improve the use of malaria prevention measures among this population. Appropriation of any control program by the population and their willingness to act is largely influenced by their level of understanding, attitude, and perception towards the disease and existing control measures [18, 19]. Hence, to promote the use of malaria interventions among adolescents, it is necessary to understand their knowledge, attitudes, and perceptions of the disease. Knowledge, attitudes, and practices (KAP) surveys are common tools that can capture critical information to guide the design of control interventions, in order to ensure community involvement, acceptance, and adherence [20, 21]. Malaria KAP surveys have been conducted in many African countries [19, 22–30]; however, to our knowledge, there is a scarcity of data on malaria knowledge, attitudes, and practices among adolescents in Senegal, specifically those living in rural areas with high malaria transmission. Therefore, the present study sought to assess malaria related knowledge (causes, symptoms, prevention methods), attitudes towards the disease, uptake of control and prevention practices and care-seeking behaviors among adolescents living in areas with persistent malaria transmission in Senegal. In addition, the study explored factors associated with adolescent knowledge, attitudes, and practices towards malaria. The main objective of this study was to generate evidence that can guide the NMCP to design social and behavior change interventions tailored to adolescents, in order to raise their awareness and improve their uptake of malaria control and prevention measures.

## Methods

### Study area and design

A community-based cross-sectional survey was conducted in the health district of Saraya in November 2021. The health district of Saraya is located in the Kedougou region in the southeastern part of Senegal, 800 km from the capital, Dakar. The district shares a border with Mali to the east, Guinea to the south, the region of Tambacounda to the north, and the health district of Kedougou to the west. The health district occupies a land area of 6,837 sq km. The population was estimated to be around 66,017 inhabitants in 2021 [31] and the district has 1 health center, 22 health posts, 28 health huts, and 78 villages with a village volunteer for malaria case management ("DSDOM", *Dispensateur de soins à domicile*). The climate is Sudano-Guinean with a dry season and a rainy season. Malaria is meso-endemic and stable in Saraya, with a long transmission season lasting 4 to 6 months from July to December. Transmission intensity remains high during the rainy season, with a high biting rate (25 bites/person/night) [32] and high morbidity during the transmission period. The major vectors are *Anopheles gambiae*, *An. arabiensis sl*, *An. funestus* and *An. nili* [33]. Malaria incidence (per 1000 population at risk) has steadily increased from 379.8 in 2019, to 607.6 in 2020, and up to 744.7 in 2021 [13, 14].

### Study population and sampling technique

A multistage random sampling technique was used to select study participants. We first selected the four health posts (out of 22 total) that reported the highest number of malaria cases in 2020 (Diakhaling, Khossanto, Mamakhono, Sambrambougou) in the Saraya health district.

In the second stage, thirty clusters, representing fifteen villages in the catchment areas of the targeted health posts, were selected using a random sampling process with probability proportional to the village's population size (PPS). The third stage consisted of selecting households to be surveyed in each village using the compact segment sampling method. A map of each village was drawn and then was divided into segments so that each segment included approximately 13 adolescents. Segments were numbered and then one segment was randomly selected. All the households in the segment were selected for inclusion in the survey [34, 35]. Households were visited by surveyors, and in each household, all resident adolescents aged 10 to 19 years old were enrolled in the study after consenting.

## Data collection method

The study questionnaire adapted from previous studies [22, 25, 26, 36–38] was developed using the Open Data Kit (ODK) platform, and was reviewed by malaria experts and field epidemiologists for accuracy and validity. The questionnaire was first designed in French then translated into Malinké, the local language in the study area. The questionnaire included closed questions that focused on the following topics: i) socio-demographic information of the participant and the household head, ii) household assets, iii) knowledge on malaria transmission, signs and symptoms of malaria, and how to prevent malaria, iv) prevention practices, v) attitudes toward malaria and, vi) care seeking practices for malaria. The questionnaire was pretested in Saraya city, which was not one of the study sites, to check for accuracy and the questionnaire was modified based on the pilot study results. Cronbach's coefficient alpha was calculated to check the reliability of each construct of the questionnaire, including knowledge, attitude, and practices. The coefficient was respectively 0.78, 0.80, 0.58, and 0.53 for knowledge, attitude, prevention, and care seeking.

Data were collected by trained field workers, using android tablets. The study team was trained on the study protocol, the questionnaire, and on the consent process. During the data collection phase, monitoring visits were performed for data quality checks and to assess compliance with the study protocol including informed consent requirements. Eligible participants included adolescents aged 10–19 years old who had lived in the selected households for at least six months.

## Outcome variables

The main outcomes assessed in this study were: malaria knowledge, attitudes, uptake of malaria prevention and control practices, and care-seeking practices for fever.

Three items were used to assess participants' knowledge on malaria: (1) correct identification of the mode of transmission, identification of the signs and symptoms of malaria, and knowledge of methods to prevent malaria. A knowledge score was calculated by summing the participant's score across the three knowledge questions. A participant who described a mosquito bite as the mode of transmission for malaria was given a score of one, otherwise the participant was assigned a score of zero. For knowledge on malaria symptoms, a score of one was granted for each correct answer for the main symptoms of malaria: fever, headache, chills, abdominal pain, and vomiting. For knowledge of how to prevent malaria, a score of one was given to participants who correctly reported a bed net as a means to prevent malaria, and half point was given if the participant identified any of the other following prevention measures: mosquito coils, wearing long clothes, insecticide spray, weeding, or sewage disposal. Malaria knowledge was categorized into two levels using the median of the total score as cut off. Participants were classified as having a poor (below or equal to the median) or a good (above the median) level of knowledge on malaria.

To assess malaria attitudes, six items related to the population at risk of malaria, prevention, and treatment were used. The responses, based on Likert's scaling technique had five possible levels, ranging from strongly agree (score 5), agree (score 4), undecided (score 3), disagree (score 2), to strongly disagree (score 1) [25, 39]. The scale was reversed for one item (malaria can be cured without medical treatment). An attitude score was estimated by adding up each participant 'score across the six variables. Participants that scored the median of the total score or above were considered having a positive attitude, and the others as having a negative attitude.

A variable to assess the practice of the participant with regard to malaria prevention was defined considering bed net ownership and use of a bed net more than three times in a week or every night [26, 27]. Participants who own a bed net and use this more than three times in a week or every night, are categorized as having a good prevention practice, whereas bad prevention practices were considered those who do not have a bed net, and those who possess a net but do not use it at least 3 times a week.

Care-seeking practices for malaria was assessed as whether the adolescent sought care at a health facility (specifically a health center, health post, health hut, or DSDOM) for recent malaria within 24 hours at the onset of the symptoms. Adolescents who sought care at a health facility within 24 hours were considered to have good care-seeking practice, otherwise, they were considered as having bad care-seeking practice.

Other covariates assessed included socio-demographic factors of the adolescent and the household head (age, sex, education level, occupation, ethnicity, marital status, wealth index). Wealth index was calculated based on household assets (water source, type of toilet, ownership of radio, TV, bicycle, fridge, combustible) and household material characteristics (type of wall, type of floor, type of roof) using a principal component analysis [40, 41]. The index was categorized into five levels (highest, fourth, middle, second, and lowest).

## Sample size calculation

With 391 adolescents sampled, the study was powered at 90%, to detect 8% variation in knowledge, assuming a prevalence of knowledge on malaria cause, symptoms and preventive methods at 50%, with alpha at 0.05 (two sided).

## Data management and analysis

A unique identifier is attributed to each participant for confidentiality. The data collected were extracted from the server and checked for the completeness and consistencies. Cleaned data were analyzed using STATA software (Stata Corp 2016). Continuous variables were described as mean, standard deviation. Categorical variables were presented as frequency and percentage. Bivariate and multivariate logistic regressions were performed to assess factors that are related to adolescents' malaria-related knowledge, attitudes, and practices with cluster robust standard error to account for the study design. Variables with a $p < = 0.25$ in the bivariate analysis, were introduced in the multivariable regression models [36, 40]. Odds ratios of these models were derived from the final model with 95% confidence interval (CI). Akaike's Information Criteria (AIC) and Bayesian Information Criteria (BIC) were used for model selection. The significance level for the final models was set at 5% (two sided). The goodness fit of the model was assessed using Hosmer- Lemeshow test.

## Ethical considerations

The study protocol was approved by the Institutional Ethics Committee of University Cheikh Anta Diop of Dakar (CER/UCAD/AD/MSN /039/2020). The protocol was presented to the

staff of the health district to seek authorization to conduct the study. Authorization to conduct the study was also sought from community leaders in the study area. Participation in the study was voluntary. Written informed consent was sought from the parents/guardians of all adolescents aged 10–17 years old. In addition, written assent was sought from adolescents aged 15–17 years old. Furthermore, older adolescents, aged from 18–19 years old were invited to provide a written consent for their participation. No personal identifiable data was collected in the study. A unique identifier was attributed to each participant. All the data collected for the study was kept confidential, with restricted access to the study staff, only for the study purpose.

## Results

### Study participants characteristics

Overall, 391 participants, aged 10 to 19 years old [mean (SD) age = 13.8(3.1)] were enrolled in the study. Most of the participants (56.0%) were female and had (60.6%) primary school level of education. The majority of the participants belonged to the Malinke ethnic group (78.3%) and forty-eight percent (48.1%) of them were students (Table 1).

### Knowledge about malaria causes, symptoms, and prevention

Out of the 391 adolescents included in the study, 317 (81.1%) reported that they had heard of malaria and of those, 87.7% (n = 278), reported that malaria is transmitted by mosquito bites. However, some participants reported other factors as possible means of malaria transmission, including sleeping with a sick person (12.9%, n = 41), through an insect bite (5.7%, n = 18), lack of body hygiene (6.3%, n = 20). Overall, 7.3% (n = 23) of the participants did not know how malaria is transmitted. The most commonly reported symptoms of malaria by adolescents were headache (89.3%, n = 283), vomiting (52.4%, n = 166), abdominal pain (30.3%, n = 96), chills (10.4%, n = 33), fatigue (6.0%, n = 19), and loss of appetite (5.1%, n = 16).

The majority of the adolescents (84.5%, n = 268) mentioned mosquito nets as a preventive measure against malaria. Other preventive measures noted by adolescents included mosquito's coils (8.8%, n = 28), weeding (7.9%, n = 25), insecticide spray (4.7%, n = 15), antimalarial drugs (5.1%, n = 16), sewage disposal (4,7%, n = 15), wearing of long cloves (3.8%, n = 12). Six percent (n = 19) of adolescents did not know any malaria preventive measures. Approximately, one-third (34.4%, n = 109) [95%CI: 29.3%–39.8%] of adolescents had good knowledge of the cause, signs and symptoms, and preventive measures of malaria (Table 2).

Participants were mostly (87.1%, n = 276) informed about malaria by health workers, specifically by community health workers (CHWs) (53.3%, n = 169) and nurses (33.7%, n = 107). Other sources of information included school (19.8%, n = 63), radio (16.4%, n = 52), television (16.1%, n = 51), and neighborhood (community members) (12.0%, n = 38) (Fig 1).

### Attitude towards malaria

Overall, 88.0% (n = 279) of the study participants believe that everybody can contract malaria and 92.1% (n = 292) perceived the disease as deadly. The majority of the participants (85.5%, n = 271) perceived malaria as a preventable disease. Furthermore, participants indicated the necessity to be tested prior to malaria treatment initiation (86.7%, n = 275) and the necessity to complete a full course of treatment when given (84.5%, n = 268). However, 46.7% (n = 148) believed that malaria could be cured without medical treatment. Considering the total score of attitude, 59.0% (n = 187) [95% CI: 53.5%–64.3%] of the participants had a positive attitude (Table 3).

**Table 1.** Socio-demographic characteristics of the study participants.

| Variable | Category | Frequency | Percentage |
|---|---|---|---|
| **Characteristics of the participants** | | | |
| **Age (in years)** | 10–14 | 232 | 59.3% |
| | 15–19 | 159 | 40.7% |
| **Sex** | Female | 219 | 56.0% |
| | Male | 172 | 44.0% |
| **Ethnicity** | Malinke | 306 | 78.3% |
| | Pular | 62 | 15.9% |
| | Other† | 23 | 5.9% |
| **Educational level** | No education | 68 | 17.4% |
| | Koranic school | 52 | 13.3% |
| | Primary | 237 | 60.6% |
| | Secondary | 34 | 8.7% |
| **Occupation** | No occupation | 89 | 22.8% |
| | Student | 188 | 48.1% |
| | Household maid | 24 | 6.1% |
| | Farmer | 14 | 3.6% |
| | Gold digger | 45 | 11.5% |
| | Vendor | 13 | 3.3% |
| | Workers | 13 | 3.3% |
| | Other* | 5 | 1.3% |
| **Wealth index** | Highest | 78 | 19.9% |
| | Fourth | 78 | 19.9% |
| | Middle | 79 | 20.2% |
| | Second | 78 | 19.9% |
| | Lowest | 78 | 19.9% |
| **Characteristics of the household's head** | | | |
| **Age (in years)** | <29 | 34 | 8.7% |
| | 30–39 | 40 | 10.2% |
| | 40–49 | 40 | 10.2% |
| | 50–59 | 49 | 12.5% |
| | 60–69 | 40 | 10.2% |
| | > = 70 | 14 | 3.6% |
| | Missing | 174 | 44.5% |
| **Sex** | Female | 25 | 6.4% |
| | Male | 363 | 92.8% |
| | Missing | 3 | 0.7% |
| **Educational level** | No education | 164 | 41.9% |
| | Koranic school | 135 | 34.5% |
| | Primary school | 80 | 20.5% |
| | Secondary school | 9 | 2.3% |
| | Missing | 3 | 0.8% |
| **Occupation** | None | 17 | 4.3% |
| | Farmer | 169 | 43.2% |
| | Gold digger | 135 | 34.5% |
| | Vendor | 21 | 5.4% |
| | Workers | 22 | 5.6% |
| | Others** | 24 | 6.1% |

(*Continued*)

**Table 1.** (Continued)

| Variable | Category | Frequency | Percentage |
|---|---|---|---|
| | Missing | 3 | 0.8% |
| **Marital status** | Married | 382 | 97.7% |
| | Others*** | 6 | 1.5% |
| | Missing | 3 | 0.8% |

† Dialounke, Bambara, Dogon, Bosso and Mossi.

* Community health worker and apprentice.

** Commhealthealth worker, imam, head of the village, shepherd, apprentice, housewife.

*** divorced and widower.

## Practice of malaria prevention

Forty-five percent (45.4%, n = 144) of the study participants reported that their households owned bed nets. The proportion of adolescents who reported using nets was 33.1% (n = 105) and those who declared having slept under a net the previous night of the survey was 27.8% (n = 88). Out of the 105 study participants who reported bed net usage, 31.4% (n = 33) stated using the bed nets in both the dry and rainy seasons and 71.4% (n = 75) reported using a bed net every night.

Forty-five percent of adolescents (45.1%, n = 143) reported using other preventive measures, including mosquito coils (60.1%, n = 86), insecticide spraying (30.8%, n = 44), and wearing of long cloves (27.3%, n = 39). Approximately one-third, 32.8% [95%CI: 27.8%-38.2%], (n = 104) of the participants have good practices in regard to malaria prevention (Table 4).

Twenty-seven percent (27,1%, n = 39) of adolescents reported not using the bed net. The main reason indicated was that they felt too hot when sleeping under a net (54.1%, n = 20). Other reasons included the smell of the net (16.2%, n = 6), "feeling trapped in the net" (10.8%, n = 4), "there is no mosquito" (10.8%, n = 4), "there is no malaria currently" (5.4%, n = 2), "don't have access to a bed net" (5.4%, n = 2).

## Care-seeking practices

Overall, 89.3% of the study participants (n = 283) reported having had a malaria episode, and among them, 86.2% (n = 244) have sought care for this episode. The majority (79.1%, n = 193) sought care at the health post, while 23.4% (n = 57) sought care at the health hut/ DSDOM. Of those that sought care, 76.6% (n = 187) sought care within 24 hours of the onset of symptoms. Altogether, 73.8% [95%CI: 68.4%–78.6%], (n = 209) of the participants demonstrated good care seeking practice for malaria (Table 5).

## Factors associated with malaria knowledge, attitude, and practices

**Malaria causes, symptoms, and prevention knowledge.** Table A in S1 Table presents the results of the logistic regression model looking at the determinants related to adolescents' knowledge of the cause, signs and symptoms, and prevention methods for malaria. Male adolescents were 60% less likely to have good knowledge [aOR = 0.40(95%CI: 0.24–0.68)]; compared to female adolescents. The odds of having good knowledge increased with education level; the odds of having a good knowledge was 5 [aOR = 5.43(95%CI: 1.91–15.46)] and 10 times [aOR = 10.41(95%CI: 2.98–36.40)] higher among adolescents who had primary and secondary level of education, respectively, compared to those that had not received any education.

**Table 2. Participant knowledge of malaria transmission, signs and symptoms, and prevention methods.**

| Variable | Category | Frequency | Percentage |
|---|---|---|---|
| **Have you heard about malaria?** | No | 74 | 18.9% |
| | Yes | 317 | 81.1% |
| **How is malaria transmitted?** | Sleep with a sick person | 41 | 12.9% |
| | Mosquito bite | 278 | 87.7% |
| | Insect bite | 18 | 5.7% |
| | Lack of body hygiene | 20 | 6.3% |
| | Oil consummation | 1 | 0.3% |
| | Contaminated food | 8 | 2.5% |
| | Dirt | 3 | 0.9% |
| | Don't know | 23 | 7.3% |
| | Other† | 3 | 0.9% |
| **Malaria signs and symptoms** | Headache | 283 | 89.3% |
| | Abdominal pain | 96 | 30.3% |
| | Chills | 33 | 10.4% |
| | Fatigue | 19 | 6.0% |
| | Loss of appetite | 16 | 5.1% |
| | Vomiting | 166 | 52.4% |
| | Fever | 6 | 1.9% |
| | Don't know | 10 | 3.2% |
| | other* | 4 | 1.3% |
| **Malaria prevention** | Mosquito net | 268 | 84.5% |
| | Spray insecticide | 15 | 4.7% |
| | Mosquito coils | 28 | 8.8% |
| | Antimalarial drug | 16 | 5.1% |
| | Weeding | 25 | 7.9% |
| | Long cloves | 12 | 3.8% |
| | Sewage disposal | 15 | 4.7% |
| | Don't know | 19 | 6.0% |
| | Other** | 8 | 2.5% |
| **Knowledge*** | Poor | 208 | 65.6% |
| | Good | 109 | 34.4% |

† water and non-use of net.

*aches and body swelling.

** go to hospital, traditional treatment, cleanliness, cannot be prevented.

*** knowledge score calculated by summing up the score of knowledge on malaria transmission, signs and prevention, then categorized in accordance to the median of the total score.

**Attitude towards malaria.** Household wealth and education level of the participants were significantly associated with participants' attitudes towards malaria. Compared to adolescents from households in the lowest wealth quintile, those from households in the highest wealth quintile were 3.49 times [aOR = 3.49(95%CI: 1.48–8.28)] more likely to have positive attitude towards malaria. The odds of positive attitude regarding malaria was respectively 86% [aOR = 0.14(95%CI: 0.36–0.55)] and 76% [aOR = 0.24(95%CI: 0.70–0.79)] lower among participants with koranic and primary education, compared to those with secondary education level (Table B in S1 Table).

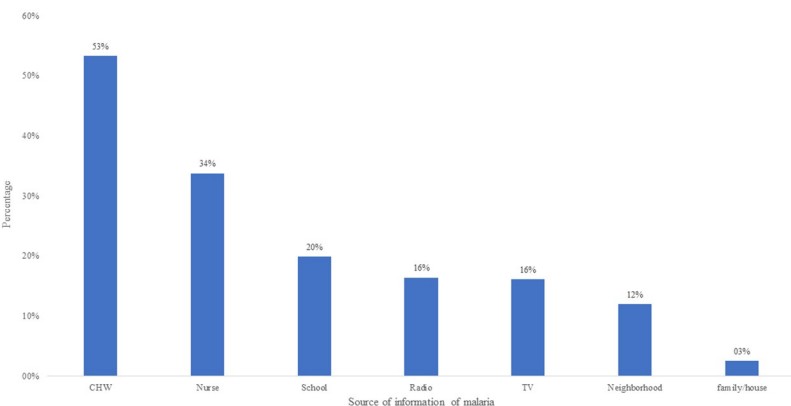

**Fig 1.**

**Malaria prevention practices.**   Adolescents who demonstrated a positive attitude towards malaria were 1.89 times [aOR = 1.89(95%CI: 1.08–3.30)] more likely to report engaging in good malaria prevention practices than those with a negative attitude. A significant association was found between household wealth and adolescent prevention practices: compared to adolescents from households in the highest wealth quintile, those from households in the fourth [aOR = 0.42(95%CI:0.19–0.89)], middle [aOR = 0.34(95%CI: 0.16–0.72)] and second wealth quintiles [aOR = 0.42(95%CI: 0.19–0.90)] were 58%, 66% and 58%, less likely, respectively, to practice good malaria prevention. Having a household head with a koranic school education was associated with good malaria prevention practice [(aOR = 1.85(95%CI: 1.04–3.28)] compared to adolescents from households where the household head received no education (Table C in S1 Table).

**Malaria care seeking behavior.**   The odds of a good care-seeking behavior for malaria was lower among adolescents from households in the second wealth quintile [aOR = 0.35(95 CI%: 0.15–0.80)] compared to those from the highest wealth quintile. In addition, the odds of good care-seeking behavior was lower among adolescents aged 15–19 years old [aOR = 0.57(95%CI: 0.32–1.02)] compared to those aged 10–14 years old, with a marginal effect (Table D in S1 Table).

**Table 3. Attitudes of the study participants towards malaria.**

| Variable | Totally agree | | Agree | | Neutral | | Disagree | | Totally disagree | |
|---|---|---|---|---|---|---|---|---|---|---|
| | N | % | N | % | N | % | N | % | N | % |
| **Everybody can have malaria** | 214 | 67.5% | 65 | 20.5% | 13 | 4.1% | 19 | 6.0% | 6 | 1.9% |
| **Malaria is deadly** | 197 | 62.1% | 95 | 30.0% | 16 | 5.1% | 5 | 1.6% | 4 | 1.3% |
| **Malaria can be cured without medical treatment** | 115 | 36.3% | 33 | 10.4% | 42 | 13.3% | 87 | 27.4% | 40 | 12.6% |
| **Malaria can be prevented** | 141 | 44.5% | 130 | 41.0% | 23 | 7.3% | 14 | 4.4% | 9 | 2.8% |
| **It is important to be tested before taking malaria treatment** | 121 | 38.2% | 154 | 48.6% | 34 | 10.7% | 5 | 1.6% | 3 | 0.9% |
| **It is necessary to finish malaria treatment** | 123 | 38.8% | 145 | 45.7% | 41 | 12.9% | 5 | 1.6% | 3 | 0.9% |
| **Attitude**[*] | | | | | | | | | | |
| **Positive** | | | | | | | | | 187 (59.0%) | |
| **Negative** | | | | | | | | | 130 (41.0%) | |

[*]calculated by summing up the score across the items used to measure attitude and categorized based on the median of the total score.

**Table 4. Malaria prevention practices of the study participants.**

| Variable | Category | Frequency | Percentage |
|---|---|---|---|
| Household owns a bed net (n = 317) | Yes | 144 | 45.4% |
| | No | 173 | 54.6% |
| Bed net use (n = 317) | Yes | 105 | 33.1% |
| | No | 212 | 66.9% |
| Bed net used the previous night (n = 317) | Yes | 88 | 27.8% |
| | No | 229 | 72.2% |
| Season of bed use (n = 105) | Rainy season | 71 | 67.6% |
| | Dry season | 1 | 0.9% |
| | All seasons | 33 | 31.4% |
| Frequency of bed net use (n = 105) | Less than 3 times per week | 1 | 0.9% |
| | 3–6 times per week | 29 | 27.6% |
| | Every night | 75 | 71.4% |
| Use other measures? (n = 317) | Yes | 143 | 45.1% |
| | No | 174 | 54.9% |
| Other preventives measures used (n = 143) | Mosquito coils | 86 | 60.1% |
| | Insecticide spray | 44 | 30.8% |
| | Grass cleaning | 2 | 1.4% |
| | Sewage cleaning | 1 | 0.7% |
| | Long cloves | 39 | 27.3% |
| | Hand cream | 10 | 3.2% |
| | Other* | 6 | 1.9% |
| Prevention practice level** (n = 317) | Good | 104 | 32.8% |
| | Bad | 213 | 67.2% |

* Topical repellent, traditional treatment, blanket, shea oil, personal hygiene.

** defined based on bed net ownership and usage.

n = number of respondents.

## Discussion

Many studies have been conducted to examine malaria control and prevention knowledge, attitudes, and practices; however, most studies have focused exclusively on adults [24, 26–29, 36, 39, 42]. Adolescents, an emerging high-risk group, have received limited attention. To address this evidence gap, our study sought to assess malaria knowledge, attitudes, and practices and key determinants among adolescents living in an area of high malaria transmission in southern Senegal.

In general, most of the adolescents demonstrated awareness of malaria transmission, symptoms, and prevention methods. This result corroborates those reported by other studies conducted in Tanzania [22], in Nigeria [38, 43–45], in India [46], and in Ethiopia [47]. The study participants reported CHWs and nurses as their main sources of malaria-related information, which was a similar finding to a study conducted in Nigeria among school adolescents [38]. This result could be explained by the fact that nurses and CHWs are often the primary points of contact for adolescent health care services in many remote settings, including in Senegal. Consequently, health care providers, such as nurses and CHWs, are well positioned to support implementation of malaria awareness strategies targeting adolescents. Conversely, in other studies conducted in Ethiopia [47] and India [46] among school-going adolescents, radio and television were reported as the main sources of information on malaria.

**Table 5. Malaria care seeking practice among the study participants.**

| Variable | Category | Frequency | Percentage |
|---|---|---|---|
| **Have you ever experienced malaria? (n = 317)** | Yes | 283 | 89.3% |
| | No | 34 | 10.7% |
| **Have you sought care? (n = 283)** | Yes | 244 | 86.2% |
| | No | 39 | 13.8% |
| **Where did you seek care? (n = 244)** | Health post | 193 | 79.1% |
| | Health center | 3 | 1.2% |
| | Health hut/DSDOM | 57 | 23.4% |
| | Self-medication | 3 | 1.2% |
| **Timeliness of care seeking after disease onset (= 244)** | The same day | 187 | 76.6% |
| | One day after | 24 | 9.8% |
| | Two days after | 29 | 11.9% |
| | More than 2 days | 2 | 0.8% |
| | Don't know | 2 | 0.8% |
| **Care seeking practice level* (n = 283)** | Good | 209 | 73.8% |
| | Bad | 74 | 26.1% |

*estimated based on care seeking in a health facility within 24 hours.

n = number of respondents.

However, most of the adolescents did not report fever as a significant symptom, and 6–7% lacked knowledge on malaria transmission and prevention. This result highlights the need for improved health-related information among this group of population. Engaging and empowering adolescents to disseminate malaria-related information in order to sensitize their peers could be an approach to fill this gap [48]. School and community-based health education programs could be used to reach this goal. This will require a multisectoral approach involving different ministries and partners such as ministry of health, ministry of education, implementing partners, as it is the case for other disease program [49].

Overall, the level of malaria knowledge regarding the cause, symptoms, and prevention is poor for the majority of the participants. The present result is similar to the findings of a study conducted in Nigeria among in-school adolescents [43]. This finding is surprising, as most of the adolescents had a primary or secondary level of education and information on malaria and how to prevent transmission are integrated into the school curriculum in Senegal. In the current study, males were significantly less likely to possess a good knowledge of malaria compared to their female counterparts. Adolescents' malaria knowledge was positively associated with their education level, with adolescents who possessed a primary and secondary level of education 5 and 10 times more likely to have a good malaria knowledge, compared adolescents who had received no education. This could be explained by the fact that educated people are more likely to access health information and have additional opportunities to be reached by messages on malaria control and prevention [50, 51].

In terms of attitude, most of the participants agreed that everyone is at risk of malaria and perceived the disease to be deadly. In addition, the majority of adolescents believe it is important to be tested for malaria before taking a malaria treatment and to complete the treatment when given. Similar findings were shown in a study conducted in Nigeria; participants also perceived malaria as deadly [38]. In our study, most adolescents believe malaria to be a preventable disease, a result that is consistent with another study conducted in Nigeria [44]. Despite many believe malaria to be preventable, just under half of the adolescents stated that

malaria can be cured without any medical treatment. This finding justifies the important need to improve communication for social behavior change among adolescents. Adolescents' attitudes toward malaria were associated with household wealth; with generally more positive attitudes among adolescents from wealthier households, and vice-versa, more negative attitudes were associated with adolescents from poorer households. In addition, positive attitude regarding malaria decreased with increase of education level, with lower odds of attitude among adolescents with koranic and primary education, in comparison to those with secondary education level. These findings could be explained by the fact that individuals from wealthier households, and those with high education level are more likely to access health information on malaria, and therefore could better understand the disease, its prevention methods as well as its seriousness [50, 51].

Regarding malaria prevention practices, less than half of the participants reported that their household own bed nets, around a third reported bed net usage, and nearly all of those who use a bed net, reported using the net between 3 to 6 times a week, or every night. The proportion of net utilization among adolescents in this study is lower compared to that indicated in studies conducted in Tanzania [22] and in Nigeria [52], but higher than the proportion reported in India [46], in other studies carried out in Nigeria [38, 53], and in Kenya [54]. Every three years in Senegal, free Long-Lasting Insecticide Nets (LLINs) are distributed to the population through mass campaigns to achieve universal coverage of LLIN, as recommended by WHO [55]. The latest campaign was carried out in 2019 and the next campaign scheduled for 2022 has not been conducted yet by the time of this study. The low proportion of bed net ownership reported in this study could be explained by the fact that the study was conducted several years after a mass LLIN campaign where the majority of the distributed nets are likely no longer being used or functional. Indeed, while LLIN are expected to last at least 3 years, studies on LLIN durability, revealed that under routine conditions LLIN can lose effectiveness because they are physically damaged, repurposed or discarded [56, 57]. Routine distribution of LLINs is also carried out at community level and in health facilities [14]. However, in the health-based routine distribution system, LLINs are only distributed free of charge for children under five and pregnant women. The low coverage of LLINs reported in this study, highlights the need for improved coverage and access. This could be achieved by expanding the target of the routine distribution to adolescents to keep them covered between mass campaigns [48]. Other channels should also be considered by the NMCP, to maintain continuous distribution and sustain LLIN coverage [58, 59].

The study revealed an important gap between LLIN ownership and LLIN usage with 25.7% of the participants reported not using the bed net, despite they have access to one. The main reason indicated was that they felt too hot when sleeping under a net. This finding justifies the necessity to reinforce sensitization to raise awareness among adolescents on the benefits of LLINs. This could be achieved through an adolescent-friendly health education program [48].

Overall, the level of good prevention practices was low. Factors associated with good prevention practice included positive attitude, adolescent education level, as well as the education level of the household head and household wealth. The odds of having a good prevention practice was 1.89 folds higher among adolescents with a positive attitude, as compared to those with a negative attitude. This finding is consistent with those indicated in studies conducted in Ethiopia [36, 51] and could be explained by the fact that the perception of the susceptibility and the threat related to malaria, and the belief in the effectiveness of the methods to avoid the disease can motivate the decision to act and adhere to prevention measures [60, 61]. This could also be explained as an increase in knowledge and favorable attitudes of an individual towards a disease, its risk factor, its seriousness, its mechanism of prevention, will lead to an increase in the probability of practicing preventive measures [36]. Furthermore, the odds of

applying a good practice of prevention is 1.85 times higher in adolescents whose household head has a koranic education level compared to those with no education. Adolescents who belonged to households from the fourth, the middle, and the second wealth quintiles were less likely to engage in prevention practices compared to those from households in the highest wealth index. This result supports the findings of a study conducted in Nigeria, concluding that higher economic class is associated with LLIN use [44] and another one conducted in Cameroon showing that wealthier people are more likely to apply good practices of malaria prevention [50]. LLINs are distributed through mass campaign approach and this strategy has been shown to ensure equitable LLIN access [55, 62, 63]. Participants who belong to the highest wealth index may have more opportunities to access information on malaria prevention including the benefits of bed-net usage; this could increase their awareness and therefore lead to higher LLIN use compared to other wealth indexes. However, this result is not in line with findings from other studies that reported a decrease in net usage with the increase of the wealth index [64, 65].

Overall, the level of care-seeking behavior was good for most of the study participants. The majority of adolescents reported having sought care for malaria in health facilities including a health center, a health post, a health hut, or from a CHW (DSDOM), within 24 hours of the start of the symptoms. This finding is similar to those of studies conducted in Nigeria [43], in Tanzania [22], and in India [46], in which most of the adolescents reported going to the hospital for malaria treatment. Regarding the associated factors, adolescents from the second wealth quintile are less likely to have good care-seeking behavior compared to those of the highest one. Since malaria treatment is free, financial resources for the transport to get to the health facilities and to pay the consultation fees may explain this difference between individuals from the second and the highest wealth indexes. This result highlights inequity in health care access among adolescents; efforts are thus needed, mainly among those from the lowest wealth households to ensure access to care.

## Limitation of the study

The cross-sectional design of the study can lead to some bias including recall bias, and reflects information collected at a specific time point, and this information may be different in another period. However, the study has been conducted during the malaria transmission season to minimize this recall bias. In addition, the estimates could have been underestimated or overestimated due to the fact that certain information collected in the study were self-reported by the participants. For instance, the use of bed nets was self-reported and could have been overestimated by the participants for social desirability. It is shown that self-reports overestimate adherence of bed net [66, 67].

## Conclusion

The study revealed that adolescents, generally have poor levels of malaria knowledge and low uptake of malaria prevention and control interventions. Targeted interventions for high-risk adolescents are needed, that focus on improving their knowledge of the disease and effective preventive measures, and on increasing their access to health care services and LLINs.

## Supporting information

**S1 File. English version of the study questionnaire.**
(DOCX)

**S2 File. French version of the study questionnaire.**
(DOCX)

**S3 File. Malinke version of the study questionnaire.**
(DOCX)

**S1 Dataset. KAP STATA dataset.**
(DTA)

**S1 Table. Tables presenting the results of logistic regression of the factors associated with knowledge, attitude, prevention, and care-seeking behaviors of the adolescent.** Tables are ordered as follows: Table A Logistic regression of the determinants related to malaria knowledge among adolescents; Table B Logistic regression of the factors associated with attitude of the participants towards malaria; Table C Logistic regression of the factors associated with malaria prevention practice; Table D Logistic regression of factors associated with malaria care-seeking behavior.
(DOCX)

## Acknowledgments

The authors acknowledge Dr Baba Camara, Head of the District Health of Saraya, and his team who facilitated the conduct of the study in the health district, as well as the population of the study area for their participation in the study.

## Author Contributions

**Conceptualization:** Fassiatou Tairou, Roger C. K. Tine.

**Formal analysis:** Fassiatou Tairou, Roger C. K. Tine.

**Investigation:** Fassiatou Tairou.

**Methodology:** Fassiatou Tairou, Roger C. K. Tine.

**Supervision:** Fassiatou Tairou, Roger C. K. Tine.

**Validation:** Fassiatou Tairou, Roger C. K. Tine.

**Writing – original draft:** Fassiatou Tairou.

**Writing – review & editing:** Fassiatou Tairou, Saira Nawaz, Marc Christian Tahita, Samantha Herrera, Babacar Faye, Roger C. K. Tine.

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
