## [Decision Letter · Decision Letter 0]

27 Sep 2022

PONE-D-22-24296Malaria prevention knowledge, attitudes, and practices (KAP) among adolescents living in an area of persistent transmission in Senegal: Results from a cross-sectional studyPLOS ONE

Dear Dr. TAIROU,

Thank you for submitting your manuscript to PLOS ONE. After careful consideration, we feel that it has merit but does not fully meet PLOS ONE’s publication criteria as it currently stands. Therefore, we invite you to submit a revised version of the manuscript that addresses the points raised during the review process.

We look forward to receiving your revised manuscript.

Kind regards,

Sammy O. Sam-Wobo

Academic Editor

PLOS ONE

Journal Requirements:

" ext-link-type="uri" xlink:type="simple">https://journals.plos.org/plosone/s/file?id=ba62/PLOSOne_formatting_sample_title_authors_affiliations.pdf"

2. Thank you for providing an English version of your questionnaire. Please also provide the questionnaire in French and Malinké

"The study was supported by the Department of Parasitology and Mycology through funding received from the Senegalese National Malaria Control Program. The funder had no role in the study design, data collection, analysis, decision to publish, or preparation of the manuscript."

Additional Editor Comments:

The reviewers noted some gaps and raised questions which you must address going forward

Do your revisions within the stipulated period and send the revisions for further action

Reviewers' comments:

Reviewer's Responses to Questions

**Comments to the Author**

1. Is the manuscript technically sound, and do the data support the conclusions?

Reviewer #1: Yes

Reviewer #2: Yes

2. Has the statistical analysis been performed appropriately and rigorously? 

Reviewer #1: Yes

Reviewer #2: Yes

3. Have the authors made all data underlying the findings in their manuscript fully available?

Reviewer #1: Yes

Reviewer #2: No

4. Is the manuscript presented in an intelligible fashion and written in standard English?

Reviewer #1: Yes

Reviewer #2: Yes

5. Review Comments to the Author

Reviewer #1: Good and well written article.

I have few questions to the authors.

1) Why this study chose adolescents as the respondents, risk of transmission are in the community as a whole regardless their age group?Despite reason study have been done among adults group not in adolescents but I think study should include the all susceptible groups rather than separated agegroup.

2) Would you please explain on validity and reliability of the questionnaires used for this study.

3) Would you please explain on an academic basis why you measure attitude similar to how we measure knowledge? Any references? In my opinion, scope of attitude should be measured by psychometric assessment. There should be NO zero mark for the answer. Please kindly refer to the concept of Likert scale. Attitude questions were so superficial, but not much can be done as data already collected.

4) Please kindly explain the basis for sample size calculation. The formula you produced is valid if your objective is to determine prevalence of KAP level; but, if you need to have bivariate and multivariate analysis, some consideration of the power of the study should be done. Hope you can revise it, with a more appropriate formula for sample size.

5) Please check the presentation of your statistical data. As example in page 11 line 256, "Overall, 391 participants, aged 10 to 19 years old (mean age= 13. 8± 3.1) were enrolled in the..." should be written as Overall, 391 participants, aged 10 to 19 years old [mean (SD) age= 13. 8(3.1)] were enrolled in the...". 

In terms of discussion, please look into the reality in the field rather than detail of statistical findings. I would suggest if we can internalize detail if we can view it in general, what does this research tell us for us to do in detail therefore allow us preventing further transmission in the community. If you think schools are useful, how we can collaborate with all agencies to make this a reality .

Reviewer #2: This is an interesting and well written piece of work.

I have just a few minor comments.

1. Lines 202 - 206. It is stated that adolescents who did not seek care for malaria at a health facility were considered to have bad care-seeking practice. Was consideration given for adolescents who may have had limited access due to a variety of reasons. For example, requiring the permission of an adult, financial constraints, off hours' time at their local health facility etc?

2. A non-response rate of 8% was considered in calculating the sample size of the study. How was this figure arrived at?

3. The results section is rather lengthy.

4. The variables on tables 4 and 5 have numbers written in bracket beside them? What do these numbers represent? If they are not required here, they should be deleted.

5. The first line of the discussion (Lines 392-393) require references for the many studies the authors refer to.

6. Were the respondents given gifts for participating in the study? The limitation described on line 514 (under-reporting of bed nets if the respondents thought they would be given nets) should not have arisen if the respondents were not told they would be given nets.

6. PLOS authors have the option to publish the peer review history of their article (what does this mean?). If published, this will include your full peer review and any attached files.

Reviewer #1: No

Reviewer #2: No

---

## [Author Response · Author response to Decision Letter 0]

10 Nov 2022

To PLOS ONE editor

Subject: Response to Reviewers

Ref: PONE-D-22-24296

Malaria prevention knowledge, attitudes, and practices (KAP) among adolescents living in an area of persistent transmission in Senegal: Results from a cross-sectional study

Dear Editor;

On behalf of the co-authors, I would like to thank the reviewers for the comments and submit the point-by-point responses to the concerns raised during the review process below:

https://journals.plos.org/plosone/s/file?id=ba62/PLOSOne_formatting_sample_title_authors_affiliations.pdf"

Authors’ reply: The manuscript has been revised in accordance with PLOS ONE’s style requirements, including those for file naming

2. Thank you for providing an English version of your questionnaire. Please also provide the questionnaire in French and Malinké

Authors’reply: The French and the Malinké version of the questionnaire are submitted with the revised manuscript

"The study was supported by the Department of Parasitology and Mycology through funding received from the Senegalese National Malaria Control Program. The funder had no role in the study design, data collection, analysis, decision to publish, or preparation of the manuscript."

Authors’reply: The study was supported by the Senegalese National Malaria Control Program, through an agreement with the Department of Parasitology and Mycology of University Cheikh Anta Diop of Dakar (# funding number: NFM 2-SEN-M-PNLP 2018 - 2019 -2020). 

Authors’reply: The funder had no role in the study design, data collection, and analysis, decision to publish, or preparation of the manuscript.

c) If any authors received a salary from any of your funders, please state which authors and which funders

Authors’reply: No one of the authors received a salary from the funder

Authors’reply: The source of funding is mentioned above (see a)

Authors’reply: The amended statements have been included in the cover letter.

Authors’reply: No retracted reference was included in the manuscript. However, to address some of the reviewer’s comments, the following references have been added to the revised version (line 633 and 638): 

Volkman HR, Walz EJ, Wanduragala D, Schiffman E, Frosch A, Alpern JD, et al. Barriers to malaria prevention among immigrant travelers in the United States who visit friends and relatives in sub-Saharan Africa: A cross-sectional, multi-setting survey of knowledge, attitudes, and practices. Carvalho LH, editor. PLoS ONE. 2020;15: e0229565

Munisi DZ, Nyundo AA, Mpondo BC. Knowledge, attitude and practice towards malaria among symptomatic patients attending Tumbi Referral Hospital: A cross-sectional study. Carvalho LH, editor. PLoS ONE. 2019;14: e0220501. 

Reviewer #1: 

1) Why this study chose adolescents as the respondents, risk of transmission are in the community as a whole regardless their age group? Despite reason study have been done among adults group not in adolescents but I think study should include the all susceptible groups rather than separated agegroup.

Authors’reply: We would like to thank the reviewer for the comment. The purpose of the current study is to generate evidence that will inform future programming and delivery of interventions to further reduce the malaria burden among adolescents. Routine data in Senegal as well as epidemiological studies have shown high performance in malaria control among the general population [1,2] but the disease burden remained high in some specific age groups such as adolescent [3]. These data call for an urgent need to design specific adolescent-oriented programs. This can only be done if malaria determinants among this specific population are well documented. Therefore, this study was designed to generate data that will contribute to filling the evidence gaps related to malaria determinants among new emerging high-risk populations including adolescents. 

2) Would you please explain on validity and reliability of the questionnaires used for this study.

Authors’reply: The study questionnaire was adapted from previous studies [4–9], and was then reviewed by malaria experts and field epidemiologists for accuracy and validity. We have calculated Cronbach’s coefficient alpha to check the reliability of each construct of the questionnaire, including knowledge, attitude, and practices. The coefficient was respectively 0.78, 0.80, 0.58, and 0.53 for questions representing knowledge, attitude, prevention, and care seeking.

3) Would you please explain on an academic basis why you measure attitude similar to how we measure knowledge? Any references? In my opinion, scope of attitude should be measured by psychometric assessment. There should be NO zero mark for the answer. Please kindly refer to the concept of Likert scale. Attitude questions were so superficial, but not much can be done as data already collected.

Authors’reply: This comment is well acknowledged; consequently, we have executed the analysis to correct this inconsistency. In the methodology section, the description of outcomes variables has been revised in accordance (line 193-200), as well as the results related to attitude (line 46, line 50-57 in the abstract section, line 298-300 in the results section). The parameters significantly associated with attitude in adolescents were also slightly changed (352-358 in the results section). These results were also interpreted and explained in the discussion (line 427- 435 in the discussion section).

The questions related to attitude were adapted from previous similar studies [7,8]. However, we acknowledge that more in-depth questions would provide more accurate information regarding attitude of adolescents towards malaria.

4) Please kindly explain the basis for sample size calculation. The formula you produced is valid if your objective is to determine prevalence of KAP level; but, if you need to have bivariate and multivariate analysis, some consideration of the power of the study should be done. Hope you can revise it, with a more appropriate formula for sample size.

Authors’reply: We agree with this comment and have revised the sample size calculation using STATA software. The following section has been added to the manuscript (line 222-224): “With 391 adolescents sampled, the study was powered at 90%, to detect 8% variation in knowledge, assuming a prevalence of knowledge on malaria causes, symptoms and preventive methods at 50%, with alpha at 0.05 (two sided).

5) Please check the presentation of your statistical data. As example in page 11 line 256, "Overall, 391 participants, aged 10 to 19 years old (mean age= 13. 8± 3.1) were enrolled in the..." should be written as Overall, 391 participants, aged 10 to 19 years old [mean (SD) age= 13. 8(3.1)] were enrolled in the...". 

Authors’reply: We have revised the presentation of statistical data regarding the mean age (line 251), and along the manuscript (Please see abstract and results section)

In terms of discussion, please look into the reality in the field rather than detail of statistical findings. I would suggest if we can internalize detail if we can view it in general, what does this research tell us for us to do in detail therefore allow us preventing further transmission in the community. If you think schools are useful, how we can collaborate with all agencies to make this a reality.

Authors’reply: The discussion section has been revised to account for this comment (line 399-405)

Reviewer #2: 

1. Lines 202 - 206. It is stated that adolescents who did not seek care for malaria at a health facility were considered to have bad care-seeking practice. Was consideration given for adolescents who may have had limited access due to a variety of reasons. For example, requiring the permission of an adult, financial constraints, off hours' time at their local health facility etc?

Authors’reply: Adolescents who did not seek care for malaria at a health facility within 24 hours were considered to have bad care-seeking practices. We have not considered those who may have had limited access to health facility for variety of reasons, in the definition of “bad care-seeking practice” because in Senegal, the diagnosis and treatment of uncomplicated malaria are free of charge, and the geographic accessibility has significantly improved due to the presence in the community of “Dispensateurs de soins à domicile”, DSDOM (village volunteer trained to test for malaria with an RDT and treated with arthemeter Lumefantrine), living in the community, as well as functional health huts (“cases de santé”), managed by a community health worker, which are accessible for patients of all ages, at any time in case of fever.

2. A non-response rate of 8% was considered in calculating the sample size of the study. How was this figure arrived at?

Authors’reply: Based on previous studies conducted in the area, by the Department of medical parasitology (unpublished data), the non-response rate ranged between 8 and 10%, therefore we considered a non-response of 8% in this study.

3. The results section is rather lengthy.

Authors’reply: The authors acknowledge this comment and have revised the results section to reduce the length by combining the section of the results related to factors associated with knowledge, attitude, prevention, and care seeking behaviors of the adolescents; and present table 6, table 7, table 8 and table 9 describing the results of KAP’factors as supplementary materials (see S5 Table)

4. The variables on tables 4 and 5 have numbers written in bracket beside them? What do these numbers represent? If they are not required here, they should be deleted.

Authors’reply: The numbers written in bracket beside each variable on tables 4 and 5, represent the number of respondents. “n” has been added in bracket beside variable on these tables, and in the caption of the tables (line 319 for table 4 and line 336 for table 5).

5. The first line of the discussion (Lines 392-393) requires references for the many studies the authors refer to.

Authors’reply: The references have been added (lines 381-383)

6. Were the respondents given gifts for participating in the study? The limitation described on line 514 (under-reporting of bed nets if the respondents thought they would be given nets) should not have arisen if the respondents were not told they would be given nets.

Authors’reply: The respondents were not promised bed nets and they were not given any other gifts to participate in the study. The participation in the study was voluntary. To avoid misinterpretation, the section has been removed from the limitation

References

1. Programme National de Lutte contre le paludisme (PNLP). Bulletin Epidemiologique annuel 2021 du paludisme au Sénégal [Internet]. 2022 Mar p. 1–76. Available from: https://pnlp.sn/bulletin-epidemiologique-annuel/

2. Agence nationale de la Statistique et de la Démographie (ANSD), Programme National de Lutte contre le Paludisme (PNLP), l’Agence des Etats Unis pour le Développement international (USAID), Fond mondial et ICF. Enquête sur les Indicateurs du Paludisme au Sénégal, 2020-2021. Rockville, Maryland, USA : ANSD, PNLP, USAID, FM et ICF; 2021. 

3. Wotodjo AN, Doucoure S, Diagne N, Sarr FD, Parola P, Gaudart J, et al. Another challenge in malaria elimination efforts: the increase of malaria among adults after the implementation of long-lasting insecticide-treated nets (LLINs) in Dielmo, Senegal. Malar J. 2018;17:384. 

4. Sumari D, Dillip A, Ndume V, Mugasa JP, Gwakisa PS. Knowledge, attitudes and practices on malaria in relation to its transmission among primary school children in Bagamoyo district, Tanzania. MWJ. 2016;7:1–7. 

5. Flatie BT, Munshea A. Knowledge, Attitude, and Practice towards Malaria among People Attending Mekaneeyesus Primary Hospital, South Gondar, Northwestern Ethiopia: A Cross-Sectional Study. Magalhães LG, editor. Journal of Parasitology Research. 2021;2021:1–14. 

6. Tesfay K, Yohannes M, Mardu F, Berhe B, Negash H. Assessment of community knowledge, practice, and determinants of malaria case households in the rural area of Raya Azebo district, Northern Ethiopia, 2017. Carvalho LH, editor. PLoS ONE. 2019;14:e0222427. 

7. Fikrie A, Kayamo M, Bekele H. Malaria prevention practices and associated factors among households of Hawassa City Administration, Southern Ethiopia, 2020. Di Gennaro F, editor. PLoS ONE. 2021;16:e0250981. 

8. Munisi DZ, Nyundo AA, Mpondo BC. Knowledge, attitude and practice towards malaria among symptomatic patients attending Tumbi Referral Hospital: A cross-sectional study. Carvalho LH, editor. PLoS ONE. 2019;14:e0220501. 

9. Eko J, Osonwa O, Offiong D. Practices of Malaria Prevention among School Adolescent within Calabar Metropolis, Southern Nigeria. Journal of Sociological Research. 2013;4:241.

---

## [Editor Report · Decision Letter 1]

16 Nov 2022

Malaria prevention knowledge, attitudes, and practices (KAP) among adolescents living in an area of persistent transmission in Senegal: Results from a cross-sectional study

PONE-D-22-24296R1

Dear Dr. TAIROU,

We’re pleased to inform you that your manuscript has been judged scientifically suitable for publication and will be formally accepted for publication once it meets all outstanding technical requirements.

Kind regards,

Sammy O. Sam-Wobo

Academic Editor

PLOS ONE
---

## [Editor Report · Acceptance letter]

22 Nov 2022

PONE-D-22-24296R1 

Malaria prevention knowledge, attitudes, and practices (KAP) among adolescents living in an area of persistent transmission in Senegal: Results from a cross-sectional study 

Dear Dr. Tairou:

I'm pleased to inform you that your manuscript has been deemed suitable for publication in PLOS ONE. Congratulations! Your manuscript is now with our production department. 

Kind regards, 

on behalf of

Dr. Sammy O. Sam-Wobo 

Academic Editor

PLOS ONE